# `BenchmarkCards`: Standardized Documentation for Large Language Model Benchmarks

**Anna Sokol**
University of Notre Dame
Notre Dame, USA
asokol@nd.edu

**Elizabeth Daly**
IBM Research
Ireland
elizabeth.daly@ie.ibm.com

**Michael Hind**
IBM Research
NY, USA
hindm@us.ibm.com

**David Piorkowski**
IBM Research
NY, USA
djp@ibm.com

**Xiangliang Zhang**
University of Notre Dame
Notre Dame, USA
xzhang33@nd.edu

**Nuno Moniz**
University of Notre Dame
Notre Dame, USA
nmoniz2@nd.edu

**Nitesh V. Chawla**
University of Notre Dame
Notre Dame, USA
nchawla@nd.edu

## Abstract

Large language models (LLMs) are powerful tools capable of handling diverse tasks. Comparing and selecting appropriate LLMs for specific tasks requires systematic evaluation methods, as models exhibit varying capabilities across different domains. However, finding suitable benchmarks is difficult given the many available options. This complexity not only increases the risk of benchmark misuse and misinterpretation but also demands substantial effort from LLM users, seeking the most suitable benchmarks for their specific needs. To address these issues, we introduce `BenchmarkCards`, an intuitive and validated documentation framework that standardizes critical benchmark attributes such as objectives, methodologies, data sources, and limitations. Through user studies involving benchmark creators and users, we show that `BenchmarkCards` can simplify benchmark selection and enhance transparency, facilitating informed decision-making in evaluating LLMs.

**Data & Code:** ⬡ github.com/SokolAnn/BenchmarkCards
🤗 huggingface.co/datasets/ASokol/BenchmarkCards

## 1 Introduction

The rapid development of large language models has opened new horizons in many fields, such as translation [8, 33], programming [37, 11, 4], medicine [48, 13, 47, 52], law [9, 46, 27], or social sciences [1, 41, 39]. However, these advancements bring significant risks, including the potential for generating biased, harmful, or misleading content, eroding public trust, and facilitating malicious activities like disinformation campaigns and fraud [49, 50, 32, 25, 10, 51, 53]. These risks often surface only after deployment, underscoring the crucial need for robust pre-deployment evaluation.

While tools like Model Cards [34], FactSheets [3], and Datasheets [23] document AI models and data, there's no framework for documenting LLMs' benchmarks. In this context, a **benchmark** is defined as a combination of a dataset, evaluation metrics, and associated pre- and post-processing

39th Conference on Neural Information Processing Systems (NeurIPS 2025) Track on Datasets and Benchmarks.

steps used to assess specific aspects of LLM behavior. This lack of standard documentation makes comparing benchmarks, choosing suitable ones, and interpreting their results hard.

However, without standardized documentation of the benchmark, it becomes difficult to understand how a benchmark can be used effectively. While frameworks like Google's Croissant [2] exist for describing machine learning datasets to enhance discoverability and usability, there remains a critical gap in standardized documentation specifically for benchmarks, which require additional metadata about evaluation methodologies and metrics that datasets alone do not address. We need to clearly understand what each benchmark measures, why it was designed in a certain way, how those factors align with real-world harms, and how results should be interpreted. Indeed, `"benchmark metadata"`, such as data provenance, metrics used, and underlying assumptions, are frequently scattered, implicit, or absent in existing benchmark documentation.

This absence of standardization hinders clear communication of crucial information, including the specific risks a benchmark addresses, its objectives, its underlying assumptions, and its datasets when using the benchmark. It can be particularly problematic for high-impact applications, making it difficult for policymakers, researchers, and the public to understand the limitations and potential biases of these benchmarks and, consequently, the LLMs they evaluate. Using benchmarks correctly can be challenging, and misuse or misinterpretation can occur if key information about their methodology and limitations is not clearly communicated. For example, benchmarks like BBQ [40] and MMLU [28] have been noted for implementation difficulties, inconsistent application, and potential misuse when their results are interpreted superficially or without understanding underlying assumptions [21]. Misuse of benchmarks can lead to false assurances of model safety or performance, potentially causing serious ethical or practical repercussions.

Additionally, beyond the challenges faced by end-users and practitioners, benchmark researchers struggle to track which LLM characteristics have been thoroughly evaluated across the ecosystem and which remain under-explored. This lack of clarity may prevent the identification of critical blind spots in existing benchmarks, particularly in areas like societal impacts or ethical concerns, hindering the strategic development of new and necessary evaluation tools.

To address this critical gap, we introduce `BenchmarkCards`, a framework inspired by similar AI documentation initiatives [34, 3, 23, 15]. Our `BenchmarkCards` framework is designed as a structured documentation system that captures and communicates essential benchmark metadata across various dimensions of model assessment, including factual accuracy, toxicity, and bias detection. By standardizing how benchmark design, assumptions, metrics, and limitations are documented, `BenchmarkCards` enables researchers and practitioners to make more informed decisions about which evaluation tools best suit their specific needs. This standardization is particularly important given the proliferation of benchmarks and the complexity of matching them to appropriate evaluation contexts. To guide the development and implementation of our framework, we seek to address the following key research questions:

**RQ1:** What elements are essential for a standardized `BenchmarkCard` template to effectively describe both the targeted LLM evaluation objectives *and* the potential limitations or biases of the benchmark itself?

**RQ2:** How should `BenchmarkCards` effectively communicate the strengths and limitations of LLM benchmarks to inform user decisions regarding benchmark selection and result interpretation (e.g., scope, purpose, biases, comparison with related benchmarks)?

To answer our research questions and validate the `BenchmarkCards` framework, we conducted a pair of user studies. First, we interviewed potential benchmark users (researchers and practitioners) to understand their current processes, challenges, and information needs when searching for, selecting, and interpreting LLM benchmarks. We then presented them with example `BenchmarkCards` to assess whether the proposed structure and content effectively address their pain points, improve clarity, and support informed decisions regarding benchmark suitability and result interpretation (addressing RQ2). Second, to evaluate the template's ability to capture essential information accurately and comprehensively (addressing RQ1), we auto-generated initial `BenchmarkCards` for a diverse set of existing benchmarks based on their publications. We then engaged the original benchmark authors, soliciting their feedback on the correctness, completeness, and overall organization of the information presented in the cards corresponding to their work. Our goal is to help users judge not only what a

benchmark measures, but also where its results may not transfer to their specific use-case or model deployment scenario.

Our studies indicate that `BenchmarkCards` improve clarity in benchmark documentation, help users differentiate between similar benchmarks, and are viewed favorably by benchmark authors as a means to accurately represent their work. Preliminary feedback suggests that `BenchmarkCards` may improve clarity in benchmark documentation and help users differentiate between similar benchmarks, with benchmark authors viewing the framework as a promising means to represent their work.

Central to our vision is fostering a community-driven ecosystem. We've set up a public repository at *github.com/SokolAnn/BenchmarkCards* not just as a collection of templates, but as a collaborative hub. Here, benchmark creators can share standardized documentation while the community collectively refines both the cards and the framework itself. Users can also automatically generate benchmark cards for their own benchmark. By making good documentation easier to create and share, we hope this practice becomes a natural part of benchmark development. Furthermore, by properly cataloging and curating metadata through `BenchmarkCards`, we can identify gaps in current evaluation coverage, allowing the research community to focus their energies on areas with less coverage rather than a scenario where many benchmarks aim to capture relatively similar capabilities and risks [42].

## 2  Related Work

Recognizing the potential for unintended consequences and ethical challenges posed by AI, researchers have developed various documentation frameworks for datasets, AI models, and systems, such as Datasheets for Datasets [23], Dataset Nutritional Label [29], Data Statements for Natural Language [7], Data Cards [44], FactSheets [3], Model Cards [34], and the CLEAR Documentation Framework [12]. Our work is inspired by such research, but differs in that we are the first to focus on AI benchmarks, their creators, and users, resulting in different documentation components. While existing frameworks like Model Cards focus on documenting AI models and FactSheets and Data Cards address dataset characteristics, no standardized approach captures the complete evaluation context that benchmarks represent. `BenchmarkCards` fill this critical gap by documenting not only the underlying data but also evaluation methodologies, targeted risks, and benchmark-specific assumptions that are essential for informed benchmark selection and interpretation.

Another line of research provides useful infrastructure and tools for LLM evaluations. Unitxt Cards [6] provide a specification framework to execute LLM evaluations that includes some components that we suggest for `BenchmarkCards`. However, the focus of Unitxt cards is to enable an automated execution of these specifications, whereas the purpose of `BenchmarkCards` is for humans to understand a broader collection of benchmark characteristics beyond what is needed for their execution. LM Eval Harness [17] is a popular evaluation engine that can take Unitxt Cards (and other specifications) as input and execute these evaluations. Olmes [26] is an extension of LM Eval Harness.

Although LLM Benchmarks attempt to measure some characteristics of an LLM, there are many high-level aspects they can target, such as performance, risks, costs, and carbon footprint. Even within these broad categories, there are subcategories. To illustrate the complexity in understanding the appropriateness of using a benchmark, we summarize one point in this vast benchmark space: risk benchmarks focused on bias. Within this targeted topic, there are many varieties of benchmarks. Our contribution is different from these tools. `BenchmarkCards` standardize the human-facing descriptions of benchmarks, including their intended use, limitations, and context to aid interpretation and proper use of those evaluation results.

For example, Winogrande [45] focuses on general commonsense reasoning and pronoun resolution, Winogender [20], WinoBias [54], and Winopron [22] focus on gender bias by evaluating how models handle gender-specific pronouns and roles in sentences. WinoQueer [18] assesses biases in LGBTQ+ contexts. RealToxicityPrompts [24] identifies toxic language, especially related to targeting marginalized groups. BBQ [39] evaluates fairness in responses concerning age, physical appearance, race, ethnicity, gender identity, sexual orientation, religion, disability, and socioeconomic status. Bias NLI [16] focuses on biases arising from linguistic nuances in natural language inference tasks. StereoSet [35] measures stereotypical biases across multiple demographic groups. TrustGPT [30] evaluates various aspects of trustworthiness, including bias, toxicity, and value alignment. CrowS-Pairs [36] measures social biases in masked language models. Although these benchmarks focus on

LLM bias, they assess different subproblems within this area. Standard benchmark documentation will help users find what is more appropriate for their goals.

In addition to these LLM benchmark activities, other researchers and organizations have developed rich risk taxonomies of LLMs [43, 31, 38, 19]. We have mapped over 130 existing LLM benchmarks to these taxonomies and discovered significant coverage gaps, where numerous benchmarks target some risks (such as bias described above), but other risks, such as "the societal impact on jobs", have limited benchmark development. Having consistent Benchmark cards for each benchmark can facilitate the easily of this gap more easily, as well as enable practitioners to more consistently communicate to nontechnical stakeholders how to interpret benchmark results.

## 3   BenchmarkCards

BenchmarkCards provide a standardized structure to comprehensively document information about an LLM benchmark. The proposed set of sections, detailed below and summarized in the template (Table 1), captures key details relevant to understanding and using a benchmark effectively. Benchmark authors should complete the sections pertinent to their specific benchmark; not all sections may apply to every benchmark (e.g., a performance benchmark might not have 'Targeted Risks' or detailed 'Ethical Considerations'). This structured approach ensures critical information regarding objectives, methodology, data, assumptions, and limitations is consistently presented, facilitating informed decision-making, comparison across benchmarks, and appropriate interpretation of results (addressing RQ1 and RQ2). Next, we will detail the purpose of each section.

The proposed set of sections is intended to provide relevant details to consider but is not intended to be complete or exhaustive, and may be tailored depending on the benchmark, context, and stakeholders. Additional details may include, for example, interpretability approaches, stakeholder-relevant explanations, and privacy approaches used in benchmark development and evaluation. The structure of BenchmarkCards ensures that users have access to critical information needed to understand the benchmark's objectives, applicability, data quality, evaluation methods, potential risks, and ethical considerations. By focusing on these key areas, BenchmarkCards facilitate informed decision-making, helping users select appropriate benchmarks and accurately interpret results, thus promoting transparency and accountability in LLM risk assessment.

**Benchmark Details:** This section of a BenchmarkCard provides basic information about the benchmark, including its name, a brief overview of its purpose and scope, the type of data used (e.g., text, code, question-answer pairs), relevant application domains, the languages represented in the benchmark data, and similar benchmarks. We use "similar" to mean a proper match between a model's goal and the benchmark's properties, including its capability focus, modality, language, domain, task format, and evaluation metric, as well as any legal or licensing constraints. For benchmarks that are multimodal or test specific skills, include tags like Modality: Vision, Text and Capability: OCR, spatial reasoning, mathematical reasoning, etc.

**Purpose and Intended Users:** This section describes the benchmark's primary objective, intended applications, target users, and specific tasks evaluated. It clarifies appropriate and inappropriate uses, providing context and highlighting limitations and out-of-scope applications to prevent mis-interpretations and misuse. We use 'task' to mean the specific evaluation activity the benchmark is designed for (question-answering, summarization, generation, etc). Authors must explicitly state both what aspects of the capability are measured and what aspects are not covered. For instance, a toxicity benchmark should specify covered domains (e.g., social media comments in English) and excluded ones (e.g., other languages, video transcripts). Within the "Purpose and Intended Users" section, explicitly include a subsection for Out-of-Scope Uses. In 2–5 bullet points, authors should give concrete examples of inappropriate uses of the benchmark. This helps users understand the benchmark's strengths

**Data:** This section provides comprehensive information about the data used in the benchmark and the data validation process. It includes the origin of the data, the scale, and its representation. If applicable, it also details the method used for data annotation. This section is important because knowing about the data helps users assess if the benchmark suits their models and understand any data-related issues that could affect results.

| Benchmark Details | |
|---|---|
| **Name:** | The official title of the benchmark used in research literature |
| **Overview:** | A concise summary of the benchmark's purpose and scope |
| **Data Type:** | The format of data evaluated (e.g., text, images, audio, multimodal) |
| **Domains:** | Field-specific applications and contexts where it's relevant |
| **Languages:** | Languages covered by the benchmark |
| **Similar Benchmarks:** | Other benchmarks with similar objectives or methodologies |
| **Resources:** | Official repositories, papers, and documentation links |
| **Purpose & Users** | |
| **Goal:** | Primary objective and intended impact of the benchmark |
| **Audience:** | Target users (e.g., ML researchers, industry practitioners, ethicists) |
| **Tasks:** | The tasks or evaluations the benchmark is intended to assess |
| **Limitations:** | Known limitations/constraints in coverage, methodology, or applicability |
| **Out-of-Scope Uses:** | Applications or interpretations that misrepresent the benchmark |
| **Data** | |
| **Source:** | Origin and collection method of benchmark data (e.g., expert-created, web-scraped) |
| **Size:** | The size of the dataset, including the number of data points or examples |
| **Format:** | Structural composition (e.g., QA pairs, text prompts, labeled images) |
| **Annotation:** | The process used to annotate or label the dataset |
| **Methodology** | |
| **Methods:** | The evaluation techniques applied within the benchmark |
| **Metrics:** | Quantitative measures employed (e.g., accuracy, toxicity scores, bias metrics) |
| **Calculation:** | Precise formulas and procedures for computing evaluation scores |
| **Interpretation:** | Guide to understanding score ranges and their significance |
| **Baseline Results:** | Reference performance from established models or systems |
| **Validation:** | Quality assurance methods to ensure benchmark reliability |
| **Targeted Risks** | |
| **Risk Categories:** | Specific LLM risks assessed by the benchmark (generated by gpt-4o-mini) |
| **AI Risk Atlas:** | Risk from AI Risk Atlas targeted by benchmark |
| **Demographic Analysis:** | Assessment of performance disparities across population groups |
| **Potential Harm:** | Possible negative impacts if models perform poorly on this benchmark |
| **Ethical & Legal Considerations** | |
| **Privacy and Anonymity:** | Measures taken to protect individual identities in the dataset |
| **Data Licensing:** | Legal terms governing benchmark usage and redistribution |
| **Consent Procedures:** | Methods used to obtain permission for data usage |
| **Compliance with Regulations:** | Adherence to relevant standards, regulations, and ethical guidelines |

Table 1: BenchmarkCard Template

**Methodology:** This section details the methodology for using the benchmark to evaluate the LLM. It includes the evaluation techniques, the performance metrics applied, how they are computed, and guidelines for interpreting them. Additional information, such as settings and prompting strategies, may also be included. This section demonstrates how the benchmark measures model performance, helping users understand and correctly interpret the results.

Benchmarks may include a set of prompts or tasks along with expected outputs, allowing evaluation of model performance against predefined metrics. However, not all benchmarks incorporate explicit quantitative metrics or focus solely on qualitative assessments; some benchmarks emphasize human evaluations to assess aspects like coherence, relevance, or ethical considerations without predefined numerical scores. `BenchmarkCards` accommodate these variations by including sections that describe the evaluation methods, whether quantitative, qualitative, or a combination of both, ensuring comprehensive documentation of how each benchmark assesses LLM risks.

By maintaining a clear and consistent definition of benchmarks and emphasizing the inclusion of both quantitative and qualitative evaluation methods, `BenchmarkCards` provides a robust framework for documenting and assessing the multifaceted risks associated with large language models. This dual approach can lead to a more nuanced understanding of model capabilities and limitations, facilitating more informed decision-making for researchers, developers, and policymakers.

**Targeted Risks:** We use an existing risk taxonomy, selecting the IBM AI Risk Atlas [31] as a structured vocabulary, to organize the specific risks the benchmark evaluates (e.g., generation

of harmful content, propagation of stereotypes, privacy violations). This section details how the benchmark operationalizes the assessment for each identified risk and specific potential harm it aims to measure. We invite benchmark creators to document underlying assumptions explicitly, such as the demographics considered, criteria used during annotation (if any), and methods for resolving disagreements. Providing this information helps users understand the scope of the risk assessment and identify potential sources of bias, strengthening the benchmark's reliability and interpretability. This section is crucial for risk-focused benchmarks but may not apply to benchmarks solely evaluating performance or capabilities.

**Ethical and Legal Considerations:** This section covers important ethical and legal aspects related to the benchmark's development and data. This includes considerations like data subject privacy and anonymity (especially compliance with regulations like the General Data Protection Regulation (GDPR)), data licensing and usage rights, consent procedures for data collection, and institutional review board (IRB) approvals where applicable. Addressing these factors provides transparency about the benchmark's ethical grounding and adherence to legal standards. Authors should address these points if relevant, particularly if the benchmark involves sensitive data, human participants, or has potential societal implications.

## 4 Demonstrative Examples

We illustrate `BenchmarkCards` using two popular benchmarks: BBQ [40] and RealToxicityPrompts [24], presenting relevant properties in Table 2.

| Benchmark Details | | |
|---|---|---|
| **Name:** | Bias Benchmark for Question Answering (BBQ) | RealToxicityPrompts |
| **Overview:** | Evaluates how consistently model responses reflect social biases | Evaluating toxicity in text generation |
| **Data Type:** | Text (QA pairs and contexts) | Text (prompts and continuations) |
| **Domains:** | Social Bias, Fairness, QA | Toxicity Detection, Controllable Generation |
| **Languages:** | English (with extension Korean, Dutch+Spanish+Turkish) | English |
| **Similar Benchmarks:** | WINO-Bias, StereoSet, CrowS-Pairs | ToxiGen, Ethos, HateCheck |
| **Resources:** | `https://github.com/nyu-mll/BBQ` | `http://toxicdegeneration.allenai.org/` |
| **Purpose & Users** | | |
| **Goal:** | Measure social biases in QA models | Measure and mitigate toxic degeneration in LLMs |
| **Audience:** | Researchers, developers, AI ethics experts | Researchers, developers, AI safety experts |
| **Tasks:** | QA with ambiguous and disambiguated contexts | Conditional text generation |
| **Limitations:** | Limited to certain social bias categories and US English | Heavily reliant on automated toxicity detection, potential for misclassification |
| **Out-of-Scope Uses:** | Deploying models without addressing biases | Using generated text without review/filtering |
| **Data** | | |
| **Source:** | Hand-crafted templates based on social biases | OpenWebText Corpus, Reddit |
| **Size:** | 58,492 unique examples | 100,000 sentence-level prompts |
| **Format:** | Textual QA pairs with contexts | Textual prompts with toxicity scores |
| **Annotation:** | Crowdsourced validation | Toxicity scores generated by Perspective API |
| **Methodology** | | |
| **Methods:** | Evaluating responses in ambiguous and disambiguated contexts | Nucleus sampling for text generation and detoxification through pretraining and word filtering |
| **Metrics:** | Accuracy, Bias Score (sDIS, sAMB) | Expected maximum toxicity, toxicity probability |
| **Calculation:** | Accuracy = % of correct answers. Bias Score = % of non-"UNKNOWN" answers that align with social bias | Expected maximum toxicity estimated via bootstrap sampling. Toxicity probability = probability of generating toxic text at least once |
| **Interpretation:** | Higher accuracy = better performance. Higher bias score = stronger reliance on social biases | Higher scores = greater tendency to generate toxic text |
| **Baseline Results:** | UnifiedQA, RoBERTa, DeBERTaV3 results | GPT-1, GPT-2, GPT-3, CTRL, CTRL-WIKI results |
| **Validation:** | Human evaluation on MTurk | Toxicity evaluation via Perspective API |
| **Targeted Risks** | | |
| **Risk Categories:** | Biases related to age, gender, race, religion, and other protected groups in QA | Risk of generating toxic or harmful language |
| **AI Risk Atlas:** | Social bias and fairness concerns in AI systems | Content safety, harmful language generation, and model misuse risks |
| **Demographic Analysis:** | Assesses bias across demographic groups | Evaluates toxicity risk across demographic groups |
| **Potential Harm:** | Risk of perpetuating harmful stereotypes or biased predictions | Risk of generating offensive or harmful content |
| **Ethical & Legal Considerations** | | |
| **Privacy and Anonymity:** | Synthetic data, no personal information | Data from OpenWebText/Reddit; anonymization efforts made |
| **Data Licensing:** | CC-BY-4.0 | Apache-2.0 |
| **Consent Procedures:** | Not applicable (synthetic data) | Data from public sources |
| **Compliance with Regulations:** | Ensured for publicly released data | Complies with applicable data privacy regulations |

Table 2: Comparison of BBQ and RealToxicityPrompts Benchmarks

**BBQ Example:** The BBQ benchmark primarily evaluates social biases within question-answering (QA) systems. As its `BenchmarkCard` (summarized in Table 2) details in the Purpose and Intended Use and Methodology sections, it uses ambiguous QA pairs where context related to sensitive attributes (e.g., race, gender, religion) might elicit biased responses. The Data section would specify the construction process focused on these ambiguous scenarios and protected categories. Its Targeted Risks focus specifically on stereotype reinforcement and disparate outcomes in the QA setting.

**RealToxicityPrompts Example:** In contrast, RealToxicityPrompts assesses the propensity of LLMs to generate toxic language when prompted. Its `BenchmarkCard` (Table 2) would clarify in its Purpose that it targets unsafe "generative" capabilities. The Methodology would describe using automated toxicity classifiers to score model outputs based on a curated set of prompts designed to range from innocuous to potentially toxic. The Targeted Risks primarily relate to the hate speech, insults, threats, and other forms of harmful content, rather than bias in classification or QA tasks.

While both benchmarks touch upon societal harms, the structured information within their `BenchmarkCards`, as exemplified in Table 2, clearly delineates their distinct goals and approaches. The fields Purpose, Methodology, and Targeted Risks are particularly crucial here. This distinction is critical for selection: a social media company aiming to filter harmful generated outputs would find RealToxicityPrompts more relevant (based on its Methodology and Targeted Risks focused on generation and toxicity scoring), while a researcher auditing fairness in an information retrieval or QA system would select BBQ (based on its Purpose, Data, and Methodology tailored to QA bias). `BenchmarkCards` thus provide the necessary structured detail to move beyond high-level labels and enable informed benchmark selection.

`BenchmarkCards` can be helpful for researchers who want to measure harmful or offensive language in the text. For example, a social media company aiming to reduce offensive content would prioritize RealToxicityPrompts, which evaluates the likelihood of toxic language generation under various prompts. However, a researcher aiming to identify bias for different demographic groups in question-answering systems would select BBQ.

## 5 User Study

To evaluate `BenchmarkCards`, we generated 100 initial card drafts from existing benchmark papers using `gpt-4o-mini` for information extraction. Then, we ran two separate studies with those generated cards targeting the two most relevant audiences: benchmark authors and users. Authors were surveyed to verify card correctness and completeness. Users were interviewed (approved by the university's internal review board) to assess the cards' utility, understandability, and comprehensiveness for their needs. The card generation prompt is in Appendix A.1.

For the User-Focused Study, we recruited 10 benchmark users (8 men, 2 women) with first-hand experience evaluating ML or LLM models. We anonymized using numerical identifiers 1 through 10. Roles included six PhD students, two post-doctoral researchers, one senior UX designer, and one technical-sales engineer. Participants represented four countries (United States, Spain, Netherlands, Portugal) and reported a mean of 5 years of ML/AI experience, spanning the full range from novice (1–3 years) to expert (>10 years). We ran 30-45 minute, remote, semi-structured interviews with each participant. The interview was structured in two parts. In the first part, we asked participants to detail their practices when looking for, selecting, and using benchmarks, including specific information they looked for. We then asked questions to understand the pain points in searching for and understanding benchmarks. Participants were given an example `BenchmarkCard` to review in the second part of the interview. After review, we used the responses from the first part of the interview to see if the `BenchmarkCard` would have addressed the difficulties identified in the second part. Finally, we elicited feedback and opportunities for improvement. Interviews were recorded and transcribed for analysis. The transcribed data was coded to identify common practices, challenges, and recurring feedback regarding the utility and design of the `BenchmarkCards`. The full interview script is available in the Appendix A.2.

For the Author-Focused Study, we emailed the authors (the example of the email is in the Appendix) on the benchmark papers and invited them to participate in a survey to provide feedback on the correctness, organization, and overall usefulness of the `BenchmarkCard`.

## 5.1 User Study Results

We conducted two complementary studies on the proposed `BenchmarkCards`: a User-Focused Study and an Author-Focused Study. We have received extensive feedback from over 25 benchmark authors and engaged 10 participants in our user study.

**User-Focused Study:** Practitioners in various ML domains found significant value in the concept and execution of `BenchmarkCards`. A strong majority reported that such standardized documentation would improve their benchmark **selection** and **understanding** processes. For instance, P-7 lauded the **conciseness**, stating, *"I like the fact that it's very to the point... I also like the fact that you know, it's all done in like a few words. So it's easier to [scroll through and] move on to the next."* This sentiment was echoed by P-3, who, when comparing to existing resources like HuggingFace, noted, *"Very few cards... have so much details like this one,"* highlighting the potential for `BenchmarkCards` to offer more comprehensive initial insights.

Participants consistently praised the **structured** approach, with sections such as *Methodology*, *Validation*, and *Targeted Risks* being frequently cited as helpful. The **importance of data annotation** quality was underscored, with P-5 emphasizing the need for annotators with domain expertise, stating, *"I don't trust annotators from non-chemistry domain when it comes to a chemistry datasets."* **License** information was also a key early factor for consideration (P-3, P-1). P-5 specifically mentioned, *"I'll jump into the validation, because I'm really interested in how the data set is annotated. And with that, I will take a look [at] the evaluation metrics."* P-6, considering a governance perspective, found the card to *"have a lot of useful information if there's multiple bias benchmarks, that they compare the ones that they want, maybe based on language, maybe based on the number of examples."*

The potential to streamline the often laborious process of gathering information was a key positive, as P-7 articulated, *"If this was me trying to decide, the first thing I would do is read the overview... This is what the data is... That would be enough for me to decide whether this would be useful or not."* The **clarity** and **ease of navigating** standardized metadata were seen as a significant step up from sifting through disparate papers or incomplete online descriptions. P-10 found the *"similar benchmarks"* section *"the most important as we can just find some related benchmarks based on these items"*. Alongside methodology and validation, participants emphasized that key decision criteria included not only annotation quality and domain expertise (P-5) but also the benchmark's **dataset size** (P-9) and its **verifiable source and recency** (P-4).

Despite the positive reception, users provided insightful suggestions to further refine the cards. A near-universal request was for the inclusion of concrete **examples** of inputs and outputs. P-1 emphasized, *"What I would look for first is what is an example of an input and an output."* This was strongly supported by P-2: *"From this card, I expect at least one example... A data example would be much better than this,"* and P-5: *"I would appreciate if there's a sample question-answer pairs."*

Another frequently identified area for improvement was the need for explicit **comparisons between similar benchmarks**. While the "Similar Benchmarks" section was appreciated, users like P-1 wanted more than just a list, suggesting *"kind of like a table or something like that, kind of like trying to compare this one with different, with other tasks, with some other similar benchmarks."* P-8 reiterated this, stating, "*The most important information is... the size comparison between different benchmarks. And why... and like, for example, concretely, the disadvantages of previous benchmarks."*

**Author-Focused Study:** Authors largely confirmed the generated cards' accuracy and clarity. Out of the 25 responses received, 10 authors indicated all the information was correct or required only minor clarifications (e.g., updating a website link or refining risk categorization). Most authors explicitly appreciated the systematic and structured format, highlighting its usefulness in improving transparency and ease of understanding. One author noted the card was *"clear, well-organized, and serves as a useful resource,"* while another found it *"well-structured and useful to quickly comprehend the benchmark details."* Almost all authors explicitly noted the importance and value of the initiative to create standardized documentation for benchmarks. Sentiments included calling the project a *"fantastic initiative"* and a "*valuable contribution to the ML community."*

Additionally, authors offered constructive suggestions aimed at improving the cards further. Common feedback included correcting factual details such as GitHub repository links, specific dataset sizes, refining descriptions of methodologies (e.g., specific classifiers used, adversarial testing inclusion), data sources (e.g., templated prompts, adaptation from existing sets), and annotation processes (e.g., scope, human vs. model); clarifying the interpretation and scope of sections like *Limitations* or

*Targeted Risks*, with some noting generated risks felt inapplicable or needed more context; and incorporating key findings, unique aspects (like balancing for grammatical case), or the full scope of work (if a repository covered multiple papers) from the original research to provide deeper context.

**Findings.** Our user studies with benchmark users and authors confirmed the utility and effectiveness of the `BenchmarkCards` framework in addressing our research questions. Feedback from creators and users highlights how this standardized approach streamlines evaluation and builds trust in assessments. Users reported that the standardized structure improves clarity and accessibility over traditional documentation, directly addressing **RQ2**. Sections like *Methodology* and *Data* facilitated rapid understanding of the benchmark scope. This consistent format also eased benchmark comparison, a major user challenge, enabling side-by-side evaluation based on fields such as *Targeted Risks* and supporting more confident, appropriate selection. Authors largely validated the template's ability to accurately capture essential benchmark elements (**RQ1**). Across the interviews, more than half of the participants independently suggested adding concrete data examples and incorporating explicit comparisons between similar benchmarks to better support benchmark selection. While our sample size limits definitive conclusions, participants reported that the standardized structure appears to improve clarity compared to traditional documentation.

## 6 Discussion

User studies confirm that `BenchmarkCards` improve the clarity and comparability of LLM benchmarks, addressing a recognized need in the AI community, aligning with the transparency goals of prior documentation efforts. By providing a standardized structure (RQ1) for documenting details such as objectives, methodology, data, and limitations, `BenchmarkCards` facilitate more informed benchmark selection and interpretation (RQ2), contributing to responsible AI evaluation practices.

The framework helps mitigate inherent benchmark design subjectivity by requiring explicit documentation of choices and assumptions. A notable component of this transparency is the encouragement for benchmark developers to identify and discuss related existing benchmarks, clarifying the contributions of their work. `BenchmarkCards` are distributed in both Markdown and machine-readable JSON, permitting immediate integration into compliance or auditing workflows [5, 14]. For the public `BenchmarkCards` repository to be a dynamic resource, it needs an active community. Outreach to benchmark developers, evaluation platform maintainers, and the broader AI/ML community can spark contributions: new cards, template refinements, automation tools, and shared best practices. Community features such as tagging, ratings, curated collections, and partnerships with platforms like HuggingFace would further boost visibility, quality, and adoption.

Looking further, the widespread adoption of `BenchmarkCards` could lead to changes in LLM evaluation. Evaluation platforms like LM Eval Harness [17] could use the structured metadata within `BenchmarkCards` to help users select evaluation suites tailored to their needs. This could improve the efficiency of evaluations by allowing for more targeted benchmark selection, potentially reducing computational costs and associated environmental impact. A publicly accessible, version-controlled repository of `BenchmarkCards` could also serve as a resource for analysis, enabling researchers to track performance trends or identify areas with limited evaluation coverage. We are actively expanding coverage through community contributions and automated generation, prioritizing widely-used benchmarks identified through citation analysis and practitioner surveys.

Such structured documentation could also strengthen AI governance and users' processes. As regulatory scrutiny of AI systems increases [38], auditors and regulatory bodies might leverage `BenchmarkCards` to assess a model's evaluation against specific standards or compliance requirements. This could foster a more evidence-based approach to AI safety assurance and assist organizations in demonstrating due diligence.

**Limitations.** While `BenchmarkCards` offer valuable documentation benefits, several limitations merit consideration. Who maintains these cards in a decentralized benchmark ecosystem? Our public repositories (`github.com/SokolAnn/BenchmarkCards`) and (`https://huggingface.co/datasets/ASokol/BenchmarkCards`) enable community contributions, but relies on sustained participation that cannot be guaranteed. The LLM ecosystem involves diverse users, making it difficult to establish clear documentation responsibility. Though we encourage benchmark creators to adopt this framework, adoption ultimately depends on evolving community norms rather than

technical solutions. We need to mention that our the interview-based approach with a limited sample may introduce social desirability and confirmation biases. Additionally, as LLMs advance rapidly and new risks emerge, `BenchmarkCards` must evolve to capture these developments, requiring ongoing updates and community engagement. A publicly accessible, version-controlled repository of `BenchmarkCards` could also serve as a resource for analysis.

Detailed documentation of benchmark limitations could potentially be exploited by malicious actors seeking to game evaluation systems or amplify harmful outputs. To address this concern, we propose implementing notification protocols where benchmark owners are alerted when their cards are modified, and establishing review mechanisms to verify changes. Additionally, as LLMs advance rapidly and new risks emerge, `BenchmarkCards` must evolve to capture these developments, requiring ongoing updates and community engagement. We will host the cards on GitHub Pages with search and filters for capability, modality, language, risk, and license in the future. An automated service will scan new benchmark papers and open pull requests with draft cards. We will integrate the cards with IBM Risk Atlas Nexus so risks and mitigations can be browsed together.

## 7 Conclusion

This paper introduces `BenchmarkCards`, a documentation framework addressing challenges in selecting and comparing benchmarks. Our user studies with benchmark creators and users show that `BenchmarkCards` improve documentation clarity, support better decision-making, and effectively represent benchmark characteristics. This approach aims to clarify benchmark properties and facilitate comparison, allowing researchers to make more informed choices about which benchmarks best suit their needs. By structuring details about objectives, methods, data, and limitations, the framework promotes transparent evaluation. While challenges remain and widespread adoption requires community effort, `BenchmarkCards` provides a solid foundation for enhancing assessment, supporting safer and more reliable AI systems.

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

# A  Appendix

This appendix provides supplementary materials referenced in the main paper.

## A.1  Prompt

Here is the prompt used to extract structured JSON `BenchmarkCards` from benchmark publications:

*Please analyze the following document and extract key information to create a structured output. Be precise and factual - only include information clearly stated in the text. For any information not available in the document, use "N/A". You may return lists for fields that have multiple items. Do NOT make assumptions or guesses. Return your response as a valid JSON object with the following structure:*

*{schema}*

*Remember:*
*1. Only include information directly stated in the paper*
*2. Use "N/A" or empty lists rather than making assumptions*
*3. Keep the structure exact - don't add or remove fields*
*4. Format as valid JSON*

Document content:
$\{text\}$

## A.2  Interview Script for Benchmark Users

The full interview script (**IRB Protocol Approved: 25-04-9220**) used for the user-focused study is as follows:

### Introduction and Consent

Hello, thank you for taking the time to speak with me today! I'm Anna, a PhD student at the University of Notre Dame. I'm researching how benchmarks are documented, especially those used in AI and machine learning. I want to understand how we can improve benchmark documentation so it is more helpful to researchers like you.

In machine learning, a benchmark is usually a dataset or task created to test and compare different models. However, the documentation for these benchmarks is often inconsistent, and figuring out which is best for a given project can be challenging. In our research, we propose "benchmark metadata cards", standardized summaries that capture key details about a dataset or benchmark. The goal is to help researchers evaluate and choose the right benchmarks more quickly. I'd like to learn about your current workflow for selecting benchmarks, and then show you a sample card to get your thoughts on its usefulness.

Today's session will take about 30-45 minutes. First, I'd like to ask you a few questions about how you usually choose benchmarks for your work. Then, I'll share a sample benchmark card with you and ask for your feedback.

Before we begin, I'd like to confirm that you consent to participate in this interview. I plan to record our conversation so I can accurately capture your feedback. Your responses will be anonymized, and nothing you say will be linked to your identity in any reports or publications. The recording will be used for research purposes only and stored securely. During the session, I'll send you a link to the benchmark card and ask you to open it on your computer. I'd also like you to briefly share your screen so I can see what you're looking at. Before doing that, please take a moment to clear any personal or sensitive information from your screen. You are welcome to pause or end the session at any point. With that, do I have your permission to record this interview and proceed?

### Part 1: Background and Current Practices

1. Could you tell me about your current role and how long you've been working in machine learning or data science?
2. What are some common types of projects you typically work on?

3. How often do you need to search or select new benchmarks or datasets for your projects?

4. Can you walk me through your usual process for identifying or evaluating benchmarks or datasets?

5. What information do you usually look for when deciding whether to use a benchmark?

   - *Follow-up:* Where do you typically look for information?
   - *Follow-up:* Do you have a go-to resource or community for benchmark recommendations?

6. What criteria are most important to you when selecting a benchmark?

7. What challenges, if any, have you faced when searching for benchmarks or evaluation datasets?

8. What are your biggest pain points in understanding how a benchmark works?

   - *Follow-up:* What specific details are consistently hard to find?

**Part 2: Feedback on Benchmark Card**

Now, I'd like to show you a short, auto-generated 'benchmark card' designed to standardize the documentation of a dataset or benchmark.

Before we look at the card, I'll send you a link. Please take a moment now to close or minimize any sensitive content on your screen. Once you're ready, go ahead and open the link. I'll give you a couple of minutes to scroll through it and read whatever sections catch your attention. Take your time, and when you're done, I'll ask a few questions about what you noticed.

1. What was your initial impression of the card? Did anything stand out as particularly helpful?

2. How well does this card convey the key information you'd need when assessing a benchmark?

3. Is there anything crucial you think is missing? If Yes, Why?

4. Is any information redundant or unnecessary? If Yes, Why?

5. Imagine you're about to run a new experiment and considering different benchmarks. Would this card help you decide which to use? Why or why not?

6. Would having cards like this reduce your need to search across multiple sources?

7. How does this benchmark card compare to the typical resources you use?

8. What might stop you from adopting a card like this into your current workflow?

   - *Follow-up:* Could this replace some of the manual searching or reading of multiple papers you currently do?

9. Is there anything you'd change about how the information is organized (any specific sections you'd reorder, rename, or combine)?

10. Would you prefer more (or less) detail for certain parts? Which parts?

11. If you had several of these cards for different benchmarks in one place, would that help you more quickly decide which one to use?

12. Is there anything else about your experience with benchmarks, dataset documentation, or these cards that you'd like to share?

13. Do you have any questions for me about this project?

**Conclusion**

Thank you for your time and for sharing your insights. We'll be analyzing this feedback to refine the benchmark cards' design. If you'd like to stay informed about our progress, I'd be happy to follow up with you once we have updated prototypes! Thank you again for your help! Your feedback is incredibly valuable to our research. If you have any follow-up ideas or questions, feel free to reach out anytime.

**Email Template for Benchmark Creators**

Dear all,

My name is Anna Sokol, and I'm a PhD student at the University of Notre Dame working on ways to improve how benchmarks and datasets are documented in machine learning research.

One of the primary goals of this project is to increase the visibility and accessibility of high-quality benchmarks through standardized documentation. We've created "benchmark metadata cards," designed to concisely summarize key information such as intended use, data characteristics, and licensing, to help researchers quickly understand and utilize benchmarks effectively. We've automatically generated a benchmark card for your paper/benchmark titled "TITLE OF PAPER" and want to ensure that it is accurate. I'd greatly appreciate it if you could take a quick look and provide your feedback. Specifically, could you let me know:

- If any details on the card are inaccurate,
- If any important information is missing,
- If anything seems unnecessary or redundant.
- Your overall impression. Is it clear and useful?

You can view the benchmark card here: "LINK TO THE BENCHMARKCARD" Just so you know, we've considered two main types of risk: those generated by the LLM itself, and those identified using the AI Risk Atlas Methodology. As a small token of appreciation, we'd happily acknowledge your contribution in our public GitHub repository or related publication. Your insights would be invaluable for refining our approach and ensuring it accurately represents your work and meets researchers' needs. Thank you very much for your time and feedback. I really appreciate it!

