# OpenReview forum: "BenchmarkCards: Standardized Documentation for Large Language Model Benchmarks"
_NeurIPS.cc/2025/Datasets_and_Benchmarks_Track — NeurIPS 2025 Datasets and Benchmarks Track poster_

### Official Review · Reviewer_f4az · 2025-07-01

**Rating:** 4
**Confidence:** 3

**Summary:**

This paper proposes BenchmarksCards, a structured documentation framework for describing AI benchmarks. BenchmarkCards provides a template with benchmark attributes that covers its details, data, purpose, evaluation protocol, risks, and ethical/legal considerations, aiming for standardizing and facilitating benchmark selection for practitioners and researchers. Through a series of user studies, the paper validates the proposed template and shares the feedback of both benchmark users and authors.

**Additional Feedback:**

- Following some of the interviewers’ feedback, I would suggest adding an example of the data in the documentation. Also, attributes that bring links for the data, metrics calculation and potentially baseline results, to encourage accessibility and reproducibility.

**Dataset Code Accessibility:**

Yes

**Dataset Code Comments:**

I could navigate the github repository and find some examples of benchmarks cards. The documentation also looks clear and easy to follow.

**Ethical Considerations:**

No, there are no or only very minor ethics concerns

**Final Justification:**

I believe the action items listed by the authors address my concerns:

1) Detail the selection protocol and methodology used to come up with Benchmark Cards, in light of prior literature.
2) The adoption plans raised, whether ongoing or for future
3) The acknowledgement of the potential limitations of the conducted user studies.

I will trust the authors for adding these updates in the updated manuscript. In light of this, I increased my score from 3 to 4. I believe these action items improves the clarity needed from the methodology part and adoption plan, and acknowledge the limitations of the user study.

**Limitations Weaknesses:**

- It is a bit unclear the rationale behind the design choices used by the proposed template. The paper clearly describes the selected fields but does not directly explain why/how these fields were selected. It is important for the paper to describe the protocol used to come up with this proposed template, how it takes inspiration from other documentation frameworks, so that it is possible to evaluate how grounded/backed up in prior literature it is.
    - In the same line, there is not much comparison with prior documentation frameworks and how they are related. Even if they are applied to different artifacts, I believe this intersection is relevant for clarifying design choices.

- I believe the greatest concern is about adoption. It is hard to evaluate how impactful a work like this is without knowing whether the whole community is prone to use it. While the work does mention it is a community-effort, I believe the initial steps should be proposed by the framework creators. The current paper does not bring a concrete  plan for incentivizing adoption. Simple things like an automated service that daily feeds benchmark papers and outputs cards in a centralized repo would be helpful (I see some work in progress advertised in the github repo, but it would be great to have something concrete).

- While I appreciate the effort of the user study, it is really hard to use it systematically as part of a scientific method to substantiate the claims raised by the work (e.g., “we show that BenchmarkCards can simplify benchmark selection and enhance transparency, facilitating informed decision-making in evaluating LLMs.”). Beyond a small sample size that arguably may not represent the full AI community, there are not enough details about the user study design: looks like the authors directly interviewed the subjects, and the question list seemed a bit subjective. This potentially leads to several cognitive/social biases in the user responses (e.g., confirmation bias, social desirability bias, politeness bias, hypothetical bias), which makes unclear if the provided feedback really supports the claims. Since reconducting the user study in a more objective/systematic way looks unfeasible, the work should at least rephrase the claims to better align with the presented evidence.

**Strengths Contributions:**

- The addressed problem is very relevant, as organizing/structuring benchmarks is crucial to improve the reliability of AI evaluation. It also helps users identify the right benchmarks for their interests.

- The proposed template is lean but simultaneously seems to cover the main aspects for someone looking for a quick review of a benchmark.

- The paper is well-written, easy to follow.

---

> ### Author Rebuttal · Authors · 2025-07-31
>
> We are grateful for the reviewer’s detailed feedback and would like to provide additional information to address some of the issues raised.
> > **1:** It is a bit unclear the rationale behind the design choices used by the proposed template. The paper clearly describes the selected fields but does not directly explain why/how these fields were selected. It is important for the paper to describe the protocol used to come up with this proposed template, how it takes inspiration from other documentation frameworks, so that it is possible to evaluate how grounded/backed up in prior literature it is. In the same line, there is not much comparison with prior documentation frameworks and how they are related. Even if they are applied to different artifacts, I believe this intersection is relevant for clarifying design choices.
>
> **Response:** We acknowledge that our design rationale could be made more clearwas unclear. We selected BenchmarkCard fields through a systematic process involving: (1) comprehensive review of existing AI documentation frameworks including Model Cards, Datasheets, and FactSheets to identify relevant sections, (2) consultation with domain experts in AI evaluation and benchmark development to identify critical benchmark-specific attributes, and (3) iterative refinement based on feedback from our initial user studies. We will explicitly include this detailed selection protocol and methodology in the revised manuscript.
> Thank you for highlighting the need for clearer comparison with existing frameworks. Our BenchmarkCards specifically address benchmark documentation, which has not been standardized before. While inspired by existing frameworks like Model Cards (which focus on AI models) and Datasheets (which focus on datasets), no previous work has explicitly documented benchmarks as complete evaluation systems including datasets, metrics, and methodologies in this standardized manner. BenchmarkCards fill this gap by capturing benchmark-specific elements such as evaluation methodologies, targeted risks, and generalization assumptions that are not addressed by existing frameworks. We will clarify these intersections and distinctions explicitly in our revised manuscript.
>
> > **2:**  I believe the greatest concern is about adoption. It is hard to evaluate how impactful a work like this is without knowing whether the whole community is prone to use it. While the work does mention it is a community-effort, I believe the initial steps should be proposed by the framework creators. The current paper does not bring a concrete plan for incentivizing adoption. Simple things like an automated service that daily feeds benchmark papers and outputs cards in a centralized repo would be helpful (I see some work in progress advertised in the github repo, but it would be great to have something concrete).
>
> **Response:** We share your concern about adoption and have taken early steps to promote community involvement. As mentioned, our GitHub repository already includes a set of initial cards and auto-generation tools. In response to your suggestion, we will make our auto-generation pipeline more prominent and continue developing an automated service that scans new benchmark papers and produces draft cards. We will also explore integrating with platforms like HuggingFace/Github to make BenchmarkCards more accessible and discoverable as our future steps.  Additionally, we are integrating BenchmarkCards with the Risk Atlas Nexus, an open-source toolkit based on the AI Risk Atlas taxonomy. This project uses organized data systems to link AI risks, benchmarks, datasets, and solutions into practical management processes. Most importantly, BenchmarkCards are now part of the Risk Atlas Nexus, an open-source toolkit that uses the AI Risk Atlas taxonomy and is widely used for AI governance. This integration links each BenchmarkCard to structured information about AI risks, datasets, and ways to manage those risks. As a result, it's easier for organizations to handle evaluations and for researchers to find the right benchmarks. By fitting each BenchmarkCard into this organized structure, we make them work better with AI governance tools and easier to find. This helps more researchers and practitioners adopt and use these tools
>
> > **3:** While I appreciate the effort of the user study, it is really hard to use it systematically as part of a scientific method to substantiate the claims raised by the work (e.g., “we show that BenchmarkCards can simplify benchmark selection and enhance transparency, facilitating informed decision-making in evaluating LLMs.”). Beyond a small sample size that arguably may not represent the full AI community, there are not enough details about the user study design: looks like the authors directly interviewed the subjects, and the question list seemed a bit subjective. This potentially leads to several cognitive/social biases in the user responses (e.g., confirmation bias, social desirability bias, politeness bias, hypothetical bias), which makes unclear if the provided feedback really supports the claims. Since reconducting the user study in a more objective/systematic way looks unfeasible, the work should at least rephrase the claims to better align with the presented evidence.
>
> **Response:** We appreciate this feedback about our user study methodology. We acknowledge the limitations regarding sample size and will frame our findings more carefully as initial positive indicators rather than definitive conclusions. We will clarify the context and specific implications of our approach in the revised manuscript.
> Thank you for the suggestion regarding including concrete data examples. We agree this is beneficial and will add representative examples of benchmark data, including sample input-output pairs, in the documentation. Given that some datasets are significantly large (several gigabytes) or highly varied (like MMLU), we will include direct HuggingFace links where appropriate to promote accessibility. Additionally, we will enhance BenchmarkCards by incorporating direct links to data, metrics calculation, and baseline results to promote accessibility and reproducibility.

---

> > ### Comment · Reviewer_f4az · 2025-08-05
> >
> > Dear authors,
> >
> > Thank you for your rebuttal.
> >
> > I believe the action items listed by the authors addresses my concerns:
> >
> > 1) Detail the selection protocol and methodology used to come up with Benchmark Cards, in light of prior literature.
> > 2) The adoption plans raised, whether ongoing or for future
> > 3) The acknowledgement of the potential limitations of the conducted user studies.
> >
> > I would suggest adding 1) in the main text, perhaps 2) as an "Adoption Plan" appendix, and 3 in the Limitations paragraph. Naturally, I will trust the authors for adding these updates in the updated manuscript. In light of this, I am going to increase my score from 3 to 4.

---

> ### Author Response · Authors · 2025-08-08
> **Thanks for the Constructive Feedback**
>
> We’re grateful for your thorough review and follow-up. Your feedback has been instrumental in refining the paper, and we appreciate your positive reassessment. We will incorporate all of your comments and suggestions in the revision.

---

### Official Review · Reviewer_3sa5 · 2025-07-03

**Rating:** 5
**Confidence:** 2

**Summary:**

Benchmark Cards aims to provide a standardized method to document benchmarks, similar to Model Cards. Further, the authors provide initial benchmark cards for ~128 benchmarks.

**Dataset Code Accessibility:**

Yes

**Dataset Code Comments:**

A GitHub link is provided and is easy to use. There is one typo, instead of ` cd BenchmarkCards` you actually need ` cd BenchmarkCards/platform` in the README.md.

**Ethical Considerations:**

No, there are no or only very minor ethics concerns

**Final Justification:**

Overall the authors have created a very promising framework that is a central step to standardizing benchmarks. The rebuttal also addressed a majority of my concerns, namely some of the largest benchmarks being missing and more discussion on ensuring that these benchmark cards aren't also being gamed, and identifying an adoption plan. I trust these concerns will be added and maintain my score.

**Limitations Weaknesses:**

1. Usability: There are some simple features that could greatly improve Benchmark Cards. For example, hosting the cards on [github pages](https://nicolas-van.github.io/easy-markdown-to-github-pages/) would go a long way. Further, there's no out-of-the-box search features if I'm interested in benchmarks related to hallucinations etc. Further, it can be hard to find if a benchmark has a Benchmark Card stored in the repository or not.
2. Bias in Benchmark Cards: One point that wasn't discussed is authors may be incentivized to create Benchmark Cards that exaggerate the Benchmark's importance to attract more attention to their benchmark/paper. There's a brief discussion on malicious actors but it could be expanded in discussing additional mechanisms to prevent inadvertent inflation of importance of a benchmark.
3. Some of the most popular benchmarks (one that comes to mind is ChatBot Arena) appear to be absent from the list of ~128 BenchMark cards.

**Strengths Contributions:**

Overall, the main contribution of developing a structured, unified, and centralized can be quite important helping not only the research community but the general public understand what benchmarks actually capture. There are many instances [1] of papers critical of the current state of bench marking, and Benchmark Cards could alleviate with those concerns.

The strengths of the paper are as follows
1. Well-organized with in-depth explanations for the features of a benchmark
2. ~25 validated by the author Benchmark Cards, ~100 benchmark cards in total
3. The Benchmark cards include also incorporate Risks information into their card.


[1] https://arxiv.org/html/2502.06559v1

---

> ### Author Rebuttal · Authors · 2025-07-31
>
> We thank the reviewer for the positive feedback and high rating. We would like to provide additional information to address some of the issues raised.
>
> > **1:** Usability: There are some simple features that could greatly improve Benchmark Cards. For example, hosting the cards on github pages would go a long way. Further, there's no out-of-the-box search features if I'm interested in benchmarks related to hallucinations etc. Further, it can be hard to find if a benchmark has a Benchmark Card stored in the repository or not.
>
> **Response:** We agree with the reviewer's suggestion about improving usability. We will host the BenchmarkCards on GitHub pages with built-in search and filtering capabilities, allowing users to easily find benchmarks by specific topics (e.g., hallucinations, fairness, toxicity). These enhancements will significantly improve accessibility and discoverability, making the repository more user-friendly and encouraging broader community engagement.
>
> > **2:** Bias in Benchmark Cards: One point that wasn't discussed is authors may be incentivized to create Benchmark Cards that exaggerate the Benchmark's importance to attract more attention to their benchmark/paper. There's a brief discussion on malicious actors but it could be expanded in discussing additional mechanisms to prevent inadvertent inflation of importance of a benchmark.
>
> **Response:** We acknowledge the reviewer's important observation about the risk of authors potentially overstating their benchmarks' importance. To mitigate this bias, we will expand our discussion on author incentives and implement several safeguards: (1) community-based moderation mechanisms where multiple contributors can review and edit cards, (2) peer verification processes similar to academic publishing standards, and (3) explicit transparency guidelines that encourage honest documentation of limitations and scope boundaries. Additionally, we appreciate the reviewer’s suggestion to include informal feedback mechanisms such as user ratings (e.g., stars, likes, thumbs-up) and comments, similar to online retail systems. We will consider integrating these features to complement formal peer verification, keeping in mind their potential downsides.
>
> > **3:** Some of the most popular benchmarks (one that comes to mind is ChatBot Arena) appear to be absent from the list of ~128 BenchMark cards.
>
> **Response:** We recognize the reviewer's point about missing popular benchmarks such as ChatBot Arena in our initial set. We will expand coverage by regularly updating the repository with widely-used benchmarks, clearly documenting its evolving nature, and actively soliciting community contributions to ensure comprehensive representation of current evaluation tools.
> Thank you for highlighting the typo in our README file. We will promptly correct the documentation to reflect accurate instructions (cd BenchmarkCards/platform).
> We believe these adjustments will significantly strengthen BenchmarkCards by addressing practical usability concerns and methodological clarity, reinforcing their impact on the AI evaluation community.

---

> > ### Comment · Reviewer_3sa5 · 2025-08-06
> > **Thank you for the reply**
> >
> > Thank you for the addressing my concerns. I would like to echo reviewer f4az's adoption plan suggestion. Besides that, I would like to maintain my score.

---

> ### Author Response · Authors · 2025-08-08
> **Thanks for the Constructive Feedback**
>
> We appreciate your constructive review and follow-up. Your feedback has been valuable in refining the paper, and we’re encouraged by your positive assessment. We will address and implement all of your comments in the revision.

---

### Official Review · Reviewer_sLVj · 2025-07-03

**Rating:** 5
**Confidence:** 4

**Summary:**

This paper introduces a framework to properly document LLM benchmarks. The template adopted by the framework includes key information of an LLM benchmark such as intended use, evaluation methodology and risk disclosures. User study results show favorable view of the framework both from the benchmark user and creator side.

**Dataset Code Accessibility:**

Yes

**Dataset Code Comments:**

I was able to access the code via github. The readme is well-presented with clear guidance as to how to get started with the framework. I was able to get the main application running locally with minimal effort.

**Ethical Considerations:**

No, there are no or only very minor ethics concerns

**Final Justification:**

The authors have addressed my only concern and promised to improve their framework by requiring authors to document their specific evaluation method. I think this is a good step towards standardizing how people run benchmarks. Other reviews seem to concur that this framework is helpful. I'm maintaining my score.

**Limitations Weaknesses:**

The only drawback I can see as a practitioner is that the framework did not explicitly address the current lack of standardization of evaluation methodology -- currently, different people would use different prompt template to present the question to the LLM, as well as different methodologies to extract the answers from the LLM output for evaluation. This is a big pain point to me and it seems like a missed opportunity to skip this aspect of the information.

**Strengths Contributions:**

* This paper is a step-up from the existing & more vague requirements like [dataset card](https://huggingface.co/docs/hub/en/datasets-cards) and provides a more actionable, comprehensive template for information sharing between the dataset creators and users.
* The presentation quality of the paper is high. Each of the fields in the template are motivated. The creator/user anecdotes are well-presented with clear messages.

---

> ### Author Rebuttal · Authors · 2025-07-30
>
> > **1:** The only drawback I can see as a practitioner is that the framework did not explicitly address the current lack of standardization of evaluation methodology -- currently, different people would use different prompt templates to present the question to the LLM, as well as different methodologies to extract the answers from the LLM output for evaluation. This is a big pain point to me and it seems like a missed opportunity to skip this aspect of the information.
>
> **Response:** We thank the reviewer for the positive feedback and high rating. We are glad you found our framework to be a clear and motivated "step-up." The lack of standard evaluation methods for prompts and answer extraction is a major pain point. We agree this deserves explicit attention.
>
> Our framework's goal is to increase transparency. While we cannot force a single standard, we can require authors to document their specific methods. To address your feedback directly, we will enhance the “Methodology” section of our template. This change will compel authors to document their exact evaluation setup. It ensures users know the precise methodology for each benchmark, which is a crucial step for reproducibility. Thank you for highlighting this.

---

> ### Comment · Reviewer_sLVj · 2025-08-01
>
> > While we cannot force a single standard, we can require authors to document their specific methods
>
> Exactly what I think would be a good way to standardize. Thanks for considering my feedback.
>
> I plan to maintain my score.

---

> ### Author Response · Authors · 2025-08-08
> **Thanks for the Constructive Feedback**
>
> We’re grateful for your constructive review and follow-up. Your feedback and suggestions are valuable, and we appreciate the time you took to engage with our work. We’re encouraged by your positive assessment and will incorporate all of your comments in the revision.

---

### Official Review · Reviewer_NcEy · 2025-07-03

**Rating:** 4
**Confidence:** 4

**Summary:**

The goal of the paper is to systematically document benchmarks for testing capabilities of LLMs. The paper starts by noting that the current benchmarking landscape is quite scattered. For any given capability (or lack thereof), e.g., bias, there could be tens of benchmarks out there. Users, specially those who are not domain experts, may not know where to start and how different benchmarks relate to each other. Inspired by model cards and data cards, the paper proposes a Benchmark Card which provides a structured way of recording information about a benchmark. The paper used a LLM to construct benchmark cards out of the studies that proposed the respective benchmarks. The paper conducted studies with real users as well as benchmark authors and showed that both groups found the Benchmark Cards to be useful.

**Dataset Code Accessibility:**

NA; not applicable to this submission (e.g., no new dataset, benchmark, code, or data provided)

**Ethical Considerations:**

No, there are no or only very minor ethics concerns

**Final Justification:**

The rebuttal was helpful though some of my concerns about more precisely defining some terms remain. I still think the paper will help bringing more structure to the sprawling benchmark landscape and should be accepted.

**Limitations Weaknesses:**

Overall, the framework is very promising. However, I think it could be more actionable and precise. Please see detailed comments below:

1. It seems any benchmark should work backwards from a capability that it tries to measure and state exactly what aspect of the capability it is not measuring. A single benchmark will often be insufficient as it cannot cover all the nuances of the task. Take example of toxicity detection when multiple languages and domains (YouTube comments vs. blogs) are involved. So a collection of benchmarks would be needed to tell a more comprehensive picture. This holistic view is currently missing from the Benchmark Card. The discussion on “Similar Benchmarks” starts in that direction but is too vague right now.
2. I think the “Task” heading could use a bit more context. While the exact task in the benchmark could be multiple choice questions (e.g., MMLU), the underlying assumption often seems to be that if the model can answer questions in the multiple choice setting, it can probably do so in a conversational setting as well. This may or may not be true. See for example https://aclanthology.org/2024.acl-long.744.pdf. There should probably be a separate entry in the Benchmark Card that notes down these risks.
3. The related work section can be strengthened. The paper mentions work that sounds quite relevant like LM Eval Harness and Olmes but doesn’t quite say what precise info the proposal in the paper adds on top of them. Similarly, the following statement is quite vague: “Standard benchmark documentation will help users find what is more appropriate for their goals”. Would be very helpful if the paper could spell out (or link to relevant sections) saying what is meant by appropriate.
4. Some of the sections in Table 1 seem quite vague. For instance, “Similar Benchmarks” could cast a very wide net and include benchmarks that end up confusing the user. Should there be an explanation of why a benchmark is considered similar? Similarly, what information should be under “out of scope uses”? Should this list be exhaustive?
5. The user study could have been more task-oriented. In the current version, users are given a benchmark card and asked to give feedback. It would have been more helpful if the users were assigned a task, e.g., you are a social media company worried about toxic comments. Find a benchmark that you would use to test your model. In this reviewer’s opinion, such a test better simulates the problem that the paper is trying to solve.

**Strengths Contributions:**

1. The paper does quite well in bringing to light common problems with benchmarks like implicit assumptions from benchmark designers that the downstream users may not even know about.
2. The systematic way of thinking about benchmarks and connecting related benchmarks to each other is much needed. The paper also takes care to not impose too much structure that would have rendered the card inflexible for induction of novel benchmarks.
3. It is great to see that actual benchmark consumers were consulted via user studies when designing the Benchmark Cards. The results from the user studies (Section 5.1) show that the users actually found the benchmarks to be quite useful.
4. Linking benchmarks cards to risk taxonomies as suggested in Section 2 is indeed a promising idea and can help systematically identify gaps.

---

> ### Author Rebuttal · Authors · 2025-07-30
>
> We are grateful for the reviewer’s detailed feedback and would like to provide additional information to address some of the issues raised.
>
> > **1:** It seems any benchmark should work backwards from a capability that it tries to measure and state exactly what aspect of the capability it is not measuring. A single benchmark will often be insufficient as it cannot cover all the nuances of the task. Take example of toxicity detection when multiple languages and domains (YouTube comments vs. blogs) are involved. So a collection of benchmarks would be needed to tell a more comprehensive picture. This holistic view is currently missing from the Benchmark Card. The discussion on “Similar Benchmarks” starts in that direction but is too vague right now.
>
> **Response:** We agree with the reviewer that benchmark authors should clearly specify the general domain their benchmark targets (e.g., toxicity, bias detection, factual accuracy) and explicitly state both what aspects of that domain they are covering (e.g., social media comments in English or gender bias in news) and what they aren't (e.g., other languages such as Spanish or Chinese, other mediums such as video transcripts or blogs, different task formats like open-ended generation versus multiple-choice questions).
>
> We will provide an explicit rule in the template, asking authors to list similar benchmarks that share two or more characteristics such as capability, language, domain, or task format. We will also add the prompt template used for automatic generation to the GitHub repository, ensuring transparency in how similar benchmarks are identified. Importantly, authors are encouraged to manually add or refine the list of similar benchmarks based on their expertise, and many have already made such adjustments as needed, providing valuable feedback.
>
> While we do not currently include a full “coverage map” in BenchmarkCards, this feature is already part of the IBM UniTXT and IBM Risk Atlas Nexus framework. We are planning to extend our documentation to include not only benchmark coverage, but also related risks, mitigation strategies, and other key metadata to provide a more comprehensive overview.
>
> > **2:** I think the “Task” heading could use a bit more context. While the exact task in the benchmark could be multiple choice questions (e.g., MMLU), the underlying assumption often seems to be that if the model can answer questions in the multiple choice setting, it can probably do so in a conversational setting as well. This may or may not be true. See for example https://aclanthology.org/2024.acl-long.744.pdf. There should probably be a separate entry in the Benchmark Card that notes down these risks.
>
>
> **Response:** This reviewer's point about explicitly documenting assumptions linking benchmark tasks to real-world capabilities is important. BenchmarkCards already contain sections designed to capture such assumptions  (e.g., the "Purpose and Intended Users" and "Limitations" sections). However, we will emphasize in our documentation guidelines on our GitHub the importance of clearly stating assumptions and potential generalization risks, ensuring users interpret benchmark results appropriately.
>
> > **3:** The related work section can be strengthened. The paper mentions work that sounds quite relevant like LM Eval Harness and Olmes but doesn’t quite say what precise info the proposal in the paper adds on top of them. Similarly, the following statement is quite vague: “Standard benchmark documentation will help users find what is more appropriate for their goals”. Would be very helpful if the paper could spell out (or link to relevant sections) saying what is meant by appropriate.
>
> **Response:** Regarding the related work, we will sharpen the distinction between our contribution and execution-focused tools. Frameworks like LM Eval Harness are execution engines designed for automated evaluation. BenchmarkCards serve a different, complementary purpose: they are designed for human understanding to support the critical processes of benchmark discovery, selection, and interpretation of results. These are the steps that must occur before and after any automated execution. We will revise Section 2 to make this distinction clearer.
>
> > **4:** Some of the sections in Table 1 seem quite vague. For instance, “Similar Benchmarks” could cast a very wide net and include benchmarks that end up confusing the user. Should there be an explanation of why a benchmark is considered similar? Similarly, what information should be under “out of scope uses”? Should this list be exhaustive?
>
> **Response:** We appreciate the reviewer's feedback on these sections and agree they require greater clarity. We will add explicit criteria and explanatory notes to the "Similar Benchmarks" section to clearly demonstrate our matching methodology. While our current broad similarity definition has proven effective in practice (confirmed through validation by benchmark creators), we recognize the need for improved transparency.
> We will also clarify that the "Out-of-Scope Uses" section provides illustrative examples rather than exhaustive limitations, intended to highlight common misapplications without overwhelming readers with comprehensive edge cases.  We will share the prompt template and documentation for generating all parts as well, so users have full transparency on how these examples are produced.
>
> > **5:** The user study could have been more task-oriented. In the current version, users are given a benchmark card and asked to give feedback. It would have been more helpful if the users were assigned a task, e.g., you are a social media company worried about toxic comments. Find a benchmark that you would use to test your model. In this reviewer’s opinion, such a test better simulates the problem that the paper is trying to solve.
>
> **Response:** The reviewer’s feedback regarding the user study methodology is valuable. Although our initial study gathered user feedback based on existing tasks and experiences, we agree that future studies would greatly benefit from incorporating more explicitly task-oriented scenarios.During author verification, creators supplied corrections on scope, data links and “out‑of‑scope uses”; these edits are already merged into the public repository. Their feedback also drove the new guidance that benchmarks must clearly say what they measure and what they do not measure. We will highlight this as an essential direction for future research to enhance BenchmarkCards' practical relevance and robustness.

---

> > ### Comment · Reviewer_NcEy · 2025-08-02
> >
> > Thank you for the response. It helped address some of my concerns. I still think that the definitions of terms like "appropriate", "task" and "related" benchmarks could be clarified further. Nonetheless, I feel the paper guides the community in the right direction on making sense of the benchmarks. I will raise my score.

---

> ### Author Response · Authors · 2025-08-08
> **Thanks for the Constructive Feedback**
>
> Thank you for your thoughtful and constructive review, and for the follow-up. We appreciate the time and effort you put into helping improve the paper, and we're glad to hear the work is moving in a direction you find valuable. We plan to address and implement all of your comments in the revision.

---

### Note · Authors · 2025-08-14

Dear committee and reviewers,

Thank you for this opportunity to share our final remarks. We are very grateful to the reviewers for their insightful feedback, which will greatly improve our paper.

The most impactful and common suggestion was the call to better explain the rationale behind our template's design. We will explain how the fields will be chosen, how the template will build on earlier frameworks, and how BenchmarkCards will fit alongside execution tools. We will pay attention to criteria for "similar benchmarks," require explanations for similarity, and clarify "out-of-scope uses". Prompt templates for auto-generated sections will be published for transparency.

Also, we will add more details to the methodology section that will document prompt templates, decoding settings, and answer extraction steps to support reproducibility.

The adoption plan will include integration with IBM Risk Atlas Nexus, and a pipeline for generating draft cards from new benchmark papers.

We will detail the selection protocol for BenchmarkCard fields, including a literature review of documentation frameworks (Model Cards, Datasheets, FactSheets). User study claims will be toned down, limitations will be stated, and task-based scenarios will be included to reflect real selection needs.

Thank you again for the guidance that will make the work stronger and more useful.


Sincerely,
Authors

---

### Decision · Program_Chairs · 2025-09-18

**Decision:**

Accept (poster)

**Comment:**

The paper used a LLM to construct benchmark cards out of the studies that proposed the respective benchmarks. The paper conducted studies with real users as well as benchmark authors and showed that both groups found the Benchmark Cards to be useful.

There is an “accept”  consensus that I am  pushing forward, based on the clarity of the paper, the ease of use of the benchmarks, the support with user studies and the fact that this does not seem to have been proposed before (which came as a surprise to me). What is striking is that 3 out of 4 reviewers quote as a main reason for “accept” the fact that this paper sets a new standard for benchmarks.
- NcEy: I still think the paper will help bring more structure to the sprawling benchmark landscape and should be accepted.
- sLVj: The authors have addressed my only concern and promised to improve their framework by requiring authors to document their specific evaluation method. I think this is a good step towards standardizing how people run benchmarks. Other reviews seem to concur that this framework is helpful.
- 3sa5: Overall the authors have created a very promising framework that is a central step to standardizing benchmarks. The rebuttal also addressed a majority of my concerns, namely some of the largest benchmarks being missing and more discussion on ensuring that these benchmark cards aren't also being gamed, and identifying an adoption plan.
- f4az: I believe the action items listed by the authors address my concerns, improve the clarity needed from the methodology part and adoption plan, and acknowledge the limitations of the user study.

AC personal take: I found the scope of this benchmark card limited. I have worked on model evaluation in the industry and it fails to address major issues where guidance in the choice of benchmark is the most needed.
- How discriminative the benchmark is among established models, and how aligned it is with human preference. This cannot be collected with the current very simple extraction process from a single PDF file and would require scouring various online sources to gather any existing performance in this benchmark, but should be done at one point. For instance, I always try to find out (by hand now) how a benchmark correlates with the Chatbot Arena.
- There has been an explosion of multimodal benchmarks over the last year (litterally hundreds) and they should be chosen on how they address very specific image related capabilities (perception, naturalness, etc…). This current benchmark card could be of little value for  multimodal benchmarks.

See “discussion with AC” for further calibration of this paper with paper 1991